# Numerical Simulations of the Effect of the Asymmetrical Bending of the Hindwings of a Hovering *C. buqueti* Bamboo Weevil with Respect to the Aerodynamic Characteristics

**DOI:** 10.3390/mi13111995

**Published:** 2022-11-17

**Authors:** Xin Li

**Affiliations:** College of Mechanical and Electrical Engineering, Suqian University, Suqian 223800, China; nanlixin@nuaa.edu.cn

**Keywords:** wing kinematics, flapping wing MAV, 3D scanning, asymmetrical bending, CFD simulation

## Abstract

The airfoil structure and folding pattern of the hindwings of a beetle provide new transformation paths for improvements in the aerodynamic performance and structural optimization of flapping-wing flying robots. However, the explanation for the aerodynamic mechanism of the asymmetrical bending of a real beetle’s hindwings under aerodynamic loads originating from the ventral and dorsal sides is unclear. To address this gap in our understanding, a computational investigation into the aerodynamic characteristics of the flight ability of *C. buqueti* and the large folding ratio of their hindwings when hovering is carried out in this article. A three-dimensional (3D) pressure-based SST k-ω turbulence model with a biomimetic structure was used for the detailed analysis, and a refined polyhedral mesh was used for the simulations. The results show that the fluid around the hindwings forms a vortex ring consisting of a leading-edge vortex (LEV), wing-tip vortex (TV) and trailing-edge vortex (TEV). Approximately 61% of the total lift is generated during the downstroke, which may be closely related to the asymmetric bending of the hindwings when they are subjected to pressure load.

## 1. Introduction

The excellent maneuverability and continuous cruising ability exhibited by the aerial flight of insects provide great practical significance for the structural optimization design and aerodynamic performance improvement in small aircraft operating in narrow locations. With continuous upgrades and improvements in scientific research equipment in recent years, more and more insects’ flight secrets have been constantly analyzed and recognized, which promoted the rapid development of a new type of small, highly maneuverable and hovering robots, known as flapping wing micro air vehicles (MAVs) [1,2,3,4,5,6,7,8,9,10,11,12]. Although flapping wing aircraft have better aerodynamic performance and flight stability than fixed-wing aircraft, insects have complex flight patterns. The flapping kinematics of insects are difficult to decompose into independent parameters, and the flow field around the wings is inherently highly viscous and unstable. Therefore, various tools are needed to analyze their performances under the flight conditions of interest [13].

Coleopteran beetles account for 40% of the total insect population [14] and are one of the most successful flying groups. The beetle’s front wings evolved into rigid elytra with the change in living environments, which can reduce the evaporation of bodies of water, promote the folding performance of the hindwings and protect the hindwings and abdomen from damage [15,16,17]. Furthermore, the coupling effect of the forewings and hindwings with mutually independent flapping motions may cause interactions in airflow between the wings. In beetles, the hindwings are considerably longer than the elytra. This means that, although the flapping motion of the beetle operates within an in-phase flapping arrangement of the elytra and the hindwing, the elytra are always passively flapping due to the muscles at the wing root of the hindwings [18]. Studies have found that most beetles keep their elytra open during flight [6,19,20,21], which indicates that they may affect aerodynamic characteristics. The flight kinematics and aerodynamic mechanism of the beetles can provide new ideas for improvement in the aerodynamic performance of the flapping-wing MAVs and the structural optimization of the airfoil of the bionic wings. However, there are relatively few experimental methods and numerical simulation studies on the aerodynamic mechanism of beetle flight.

The aerodynamic performance of the hindwings and elytra of a beetle *Epilachna quadricollis* in hovering flight was studied by a two-dimensional computational fluid dynamics (CFDs) simulation [22]. To determine the aerodynamic function and effect on the performance of the elytra, the wake of tethered dung beetles *Heliocopris hamadryas* was explored [15]. The aerodynamics of the beetle *Trypoxylus dichotomus* were investigated by CFD analysis [19]. Numerical simulations were used to explore the influence of the microstructure of the elytra and the blood flow in hindwing’s veins on the aerodynamic force of *Coccinella septempunctata* [14]. The airflow characteristics around the elytra and the hindwings of the *T. dichotomus* beetle were observed [23]. The aerodynamic force on an elytron of rhinoceros beetle *T. Dichotomus* during flapping in forward flight was investigated by using CFD simulations and by considering elytron–hindwing interactions [20,24]. The SST k-ω turbulence model based on the three-dimensional (3D) pressure simulation was used to analyze the take-off and landing performance of the *Rhinoceros* beetle [25]. The relationships between the attack angles and the rigidizable behavior of the hindwings of the ladybird *Harmonia axyridis* were investigated by numerical simulations [26]. The design of the flapping-wing system that imitates the flapping-wing kinematics data of the *Protaetia brevitarsis* beetle lacks supporting data from the aerodynamic parameters of the CFD simulation of the beetle’s flapping wing [27]. In recent years, experiments and numerical simulations contributed greatly to addressing the aerodynamic mechanism of beetles’ flight. However, the geometric model of the bionic wings in these numerical analyses is quite different from the geometric surface shape of the real beetle’s wings, which cannot fully map the aerodynamic mechanism of the real beetle’s flapping wings. Therefore, further optimizing the inverse modeling technology of beetle wings is necessary.

The phenomenon of asymmetric bending, which occurs when equally distributed loads of the same magnitude are applied from the dorsal and ventral sides of insect wings, is found in the wings of the blowfly [28], butterflies [29], the hawk moth *Manduca sexta* [30], the hindwings of locusts [31] and the *A. dichotoma* beetle [32,33]. Insect wings typically exhibit significant asymmetric deformation patterns due to the presence of free flow that prevents the cancelling of drag forces that are generated between the downstroke and upstroke, which in turn exhibits a greater amplitude during the upstroke than during the downstroke [34]. The use of asymmetric passive deformations can increase the lift of flapping-wing MAVs and improve the power consumption of the motor [35]. The asymmetrical bending of the hindwings of the *A. dichotoma* beetle [33] was investigated, and it was found that the hindwings are affected by the asymmetrical bending of insect wings when the membranes are stiffened due to stress. However, the simulated 3D model of the hindwing is a flat wing, which is quite different from the real cambered and corrugated shape of the hindwing of the beetle. The effect of the asymmetric bending of insect wings on aerodynamic performance is still unclear. Therefore, it is necessary to reconstruct an accurate geometric surface model of the hindwings and to reveal the underlying theory behind the experimental results via numerical simulation methods. A computational investigation on the aerodynamic characteristics of the hindwings of the *C. buqueti* with asymmetric bending effects in hovering flight is carried out in this article.

## 2. Materials and Methods

### 2.1. Specimens

Adults of *C. buqueti* were obtained from the bases of bamboo plants in the city of Suqian, Jiangsu province, China. These specimens (20 beetles) have an average body length of 41.41 ± 1.12 mm, an average mass of about 4.28 ± 0.16 g and an average wing span of about 109.79 ± 0.32 mm.

### 2.2. Biological Scanning Operation

Accurate modeling techniques should be used to reconstruct the geometric surface model of *C. buqueti* to analyze the asymmetric bending of the hindwings. The chemical treatment process of the hindwings before 3D scanning is outlined in refs. [36,37]. Unlike the previous scanning method [36,37], the scanner here is orthogonal to the dorsal side of the hindwing, which is determined by the distribution of convex veins in the vein’s venation. In addition, gradual scanning is performed from the wing root to the wing tip due to the fact that many noise points will be generated at the beginning of the procedure, and point cloud data at the wing root can be greatly simplified with the inverse modeling without affecting any subsequent simulation results. To complete the symmetry modeling of the hindwings model, a pair of hindwings can be scanned for each experiment and repeated three times. In reverse modeling and 3D reconstruction, the surfaces of a pair of wings can be cut into left and right wings.

### 2.3. Definition of Euler Angles

Wing motion parameters are key factors in constructing the wing kinematics model. In order to describe the flapping kinematics of the hindwing and facilitate CFD simulation, defining the Euler angle of the wing-flapping change is necessary. In our previous work [21], we observed the flapping-wing flight of beetles via tethering experiments. A high-speed camera system was used to capture the flapping-wing characteristics of *C. buqueti* using multiple attitude adjustments. Image-processing and data-processing software were used to mark the captured video frame by frame. The stroke plane connected the wing root and wing tip at the start and end of the downstroke. The position angle, *ϕ*(t), is defined as the angle between the wing base-to-wing tip line and the projection of the wing base-to-wing tip line on the horizontal plane. The deviation angle, *θ*(t), is the angle between the wing base-to-wing tip line and the stroke plane. The angle of attack, *α*(t), is the angle between the stroke plane and the chord line.

### 2.4. CFD Numerical Method

It is an indispensable step in CFD simulations to set the size of the computational domain in a limited space. A smaller airflow simulation environment can simplify the computational task so that boundary disturbances are negligible. The boundary setting of compressible or incompressible fluids usually depends on the simulated geometric model, and a cuboid is chosen as the computational domain for the flapping flight of *C. buqueti*. The lengths of approximately 5–10 times the wing or 10–20 times the mean chord of the insect wing are often taken as references for determining the calculation domain [4,22,38]. In this study, the dimension of the computational domain is 0.3 m in length (approximately 7 times the wing length), 0.36 m in width (approximately 8 times the wing length) and 0.22 m in height (approximately 5 times the wing length). The ANSYS ICEM-CFD software was used to mesh the geometrical model.

For the insects hovering in still air, the Reynolds number (denoted by Re), which represents the ratio of inertial effects to viscous effects [2,22], was calculated to be
(1)Re=2ρcϕfRv=8519
where *ρ* is the density of air, which is 1.205 kg/m^3^; *v* represents dynamic viscosity and is 1.81 kg/m^3^; *f* represents flapping-wing frequency, which is 67 Hz [21]; *c* is the mean hindwing chord length and is 16.23 mm [39]; *R* is the length of the hindwing, which is measured to be 45.27 mm; *ϕ* is the wing beat amplitude and is about 70º [21].

In Equations (3) and (4), the Reynolds number for this hovering beetle was 851. Because the Reynolds number here is over 4000, the effect of the turbulence is not negligible in this fluid, so it is assumed that the flow field around the hovering *C. buqueti* is turbulent. Consequently, the governing equations require solutions including the three-dimensional, incompressible, unsteady continuity and shear-stress transport (SST) k–ω model for turbulent flow, as shown in their conservation forms (Equations (2) and (3)). The SST k-ω model is similar to the standard k-ω model but includes further improvements. Both the standard k-ω model and the transformed k-ε model are multiplied by the mixing function and the two models are added together. The SST model incorporates a damped cross-diffusion derivative term in the ω equation.
(2)$$\frac{\partial }{\partial t}\left(pk\right)+\frac{\partial }{\partial x_{i}}\left(pku_{i}\right)=\frac{\partial }{\partial x_{j}}\left(\Gamma_{k}\frac{\partial k}{\partial x_{j}}\right)+{\tilde{G}}_{k}-Y_{k}+S_{k}$$
(3)$$\frac{\partial }{\partial t}\left(p\omega \right)+\frac{\partial }{\partial x_{j}}\left(pku_{j}\right)=\frac{\partial }{\partial x_{j}}\left(\Gamma_{\omega }\frac{\partial \omega }{\partial x_{j}}\right)+G_{\omega }-Y_{\omega }+D_{\omega }+S_{\omega }$$

In the above equations, ${\tilde{G}}_{k}$ represents the turbulent kinetic energy due to the average velocity gradient t; $G_{k}$ is calculated from and given by
(4)$${\tilde{G}}_{k}=\min\left(G_{k},10\rho \beta^{*}k\omega \right)$$
(5)$$G_{\omega }=\alpha \frac{\omega }{k}G_{k}$$
where $G_{\omega }$ is generated by ω and is calculated as described for the standard k-*ω*; $\Gamma_{k}$ and $\Gamma_{\omega }$ represent the effective diffusion terms of k and ω, respectively; $Y_{k}$ and $Y_{\omega }$ represent the dissipation of k and ω due to the turbulence intensity; $D_{\omega }$ represents the orthogonal divergence terms; $S_{k}$ and $D_{\omega }$ are user-defined terms.

The CFD methods such as the immersed boundary method [26,27] and the overset grid method [2] have been proposed to solve the fluid dynamics problems of moving objects. In this study, a dynamic mesh method-based commercial CFD package (Fluent) was selected as the flow solver. At the initial time, both the *C. buqueti* model and the entire flow field were assumed to be stationary. For boundary conditions, the surface of the entire *C. buqueti* model as well as the outer boundary of the calculation domain were set to be a rigid wall on which a no-slip condition applied. The velocity inlet boundary condition is set at the flow field inlet of the computational domain. According to the experimental observation, the forward flight speed of the *C. buqueti* was recorded as 2 m/s [21]. Therefore, a streamlined jet with an incoming velocity of 2 m/s flowed around the *C. buqueti* model. At the outlet of the flow field, the outlet pressure’s boundary conditions were established. The relative pressure was set as 0 Pa, and the ambient atmospheric pressure was set as 101,325 Pa. The pressure-based solver was selected, and the SIMPLE algorithm was used for pressure–velocity coupling. A first-order implicit scheme was employed to quantify the discretization of simulation times. The second-order upwind scheme was used for spatial discretization by the SST k-ω model. Numerical computations converge only when all relevant residuals are below some tolerance (10^−4^ for both continuity and momentum equations).

### 2.5. Finite Element Simulation

In this study, the point force loading and pressure loading were employed to evaluate the structural performance of the hindwings of *C. buqueti*. The folding and unfolding of the hindwings rely on the bending zone and the marginal joint to realize the Z-shaped folding pattern [17]. If forces are applied to an area other than the marginal joint, the hindwing will bend unstably and fold at these positions. In addition, the wing membrane of the real hindwings is very thin and is easily damaged when point forces are applied. For pressure loads, uniformly distributed pressure is applied to the hindwing’s area. Both the point force and the pressure loads are calculated from the average vertical force obtained by CFD simulations. For material properties, the material stiffness of the wing membrane was set to 2.5 GPa and the average wing vein stiffness was set to 4.75 GPa. The material density of the hindwing’s cuticle was chosen to be 800 kg/m^3^, and the Poisson’s ratio was set to 0.25 [39]. A two-node pipe element was used to simulate the veins, and a four-node shell element (SHELL 181) was used to simulate the membrane. The quadrilateral mesh was used to mesh the wing’s membrane as it is more accurate during iterations. The wing model was fixed at the wing root with zero displacement and rotation, and the point force or pressure loading was applied to the marginal joint point from the dorsal side and ventral side. Then, inconsistencies in the deformation of the dorsal and ventral sides were compared to explain the asymmetrical bending of the hindwings of *C. buqueti*.

## 3. Results

### 3.1. Reverse Reconstruction

The point cloud data of the specimens were obtained by step-by-step scanning of the area from the wing root to the wing tip with a 3D scanner. Based on Geomagic software for inverse modeling, point cloud data were simplified, the noise points were removed and a multi-block surface sheet model of the hindwings was constructed. The surface model of the hindwings was imported into CAD software for regional thickening and surface merging. Extracting the contour curve of the hindwings and projecting it onto a reference plane, and then extruding the contour curve of the hindwing from that plane and across the reconstructed hindwing model, can split the double wing. The reverse reconstruction of the specimen’s 3D geometric model was completed, as shown in Figure 1. The 3D model of a hindwing had an average thickness of 0.004 mm [39], which is the same thickness as the real hindwings measured by scanning electron microscopy.

### 3.2. Flapping Kinematics of the Hindwings

Over 30 flapping videos have been taken, from which the one with characteristics closest to hovering flight was selected for this study. In the video frame of the stable flapping motion of hindwings transitioning from a folded configuration to an unfolded state, a wing beat cycle includes four stages: downstroke, supination, upstroke and protonation. The curved morphology of the hindwing’s surface exhibits an umbrella effect during downstroke (*t*/*T* = 0.5), as shown by the green line in Figure 2a. The position where the hindwings are at their maximum downstroke angle is at the moment of supination. The leading edge of the hindwings is almost a straight line from the wing root to the wing tip, and there is no significant bending deformation on the wing’s surface. However, the surface bending during the upstroke of the hindwings is not obvious, and there is no umbrella effect (*t*/*T* = 0.0). The position where the hindwings are at their maximum upstroke angle is at the moment of protonation.

The wing kinematics parameters are obtained from captured video frames. The time-consuming nature of the upstroke and downstroke motions is almost the same. The forward velocity was 2 m/s with zero degrees with respect to the body angle. A wing beat cycle has 30 frames, which means that the flapping frequency is about 67 Hz. The inclination angle of the flapping plane with respect to the horizontal direction is approximately 36 ± 1.3° for hovering flight. The flapping-wing amplitude (*ϕ*) of the hindwings is 70 ± 3.6°. The range of variations in the deviation angle (*θ*) is from −40° to 42°. Angles *θ*(t) and *α*(t) gradually decrease during the first half of the downstroke. The change in the motion of the flapping wing from downstroke to upstroke occurs when the downstroke of the hindwings reaches its lowest position. In the second half of the upstroke, angles *θ*(*t*) and *α*(*t*) gradually increase. At the end of the upstroke, the motion of the hindwings transitions from upstroke to downstroke. During the downstroke motion, the instantaneous angle of attack varied from 4° to 92°, which resulted in specific characteristics in the flapping-wing flight of *C. buqueti* with respect to force generation. Larger deformations occur in the flapping of the hindwings, and the full kinematics of hindwings can be found in [21].

### 3.3. CFD Simulation Results

Considering that the motion of the moving mesh will greatly increase the computational complexity of the numerical simulation, the more versatile tetrahedral unstructured mesh was used for the meshing of the computational domain. Five different grid systems and two different computational domains were calculated to analyze the effects of different grid qualities and computational domain sizes on the lift and thrust of the flapping wings. As shown in Figure 3, these grid systems are defined as a coarse mesh model, medium mesh model, fine mesh model, refined mesh model and re-refinement mesh model. In order to compare the influence of different calculation domains on the simulation results, two different computational domains (computational domain 1 and computational domain 2) were defined on the basis of a refined mesh model. After the detailed mesh refinement calculation of the tetrahedral element, the fine mesh system in the computational domain was selected as the real simulation data.

The primary simulation results are shown in Figure 4. The element size increased by 37% when the mesh was refined from coarse mesh to medium mesh. The changes in lift characteristics, from a coarse grid system to a refined grid system, manifested as a significant difference between the lift curves. The lift from the refined grid system to the re-refinement grid system does not change significantly, and the difference between the average values of the vertical forces is about 2%. Furthermore, the difference in the mean lift between computational domain 1 (time step size: 0.00001) and computational domain 2 (time step size: 0.000001) of the re-refinement grid system is less than 1%. Therefore, for the computational domain 1, the number of elements and nodes of the computing model significantly increased from the refined mesh model to the re-refinement mesh system. The difference in lift and drag from computational domain 1 to computational domain 2 is small, and the simulation model only increases the size and number of elements in the computational domain, which in turn greatly increases the calculation time. Therefore, the refined grid system in computational domain 1 was used for the simulation analysis results in this paper. The center of the computational domain was meshed into a uniform fine grid. The volume mesh and surface mesh of the computational domain were set to 0.01 m. The hindwing mesh in the inner central area of the computational domain was set to 0.0002 m. The number of elements and nodes after meshing were 4,206,849 and 748,604, respectively. The time step size in this study was 0.00001 to ensure that stable solutions were obtained throughout the simulation.

CFD simulations were conducted using the kinematics of the hindwings from a three-dimensional surface reconstruction. The vortex formation and velocity vectors at *t*/*T* = 0.3, 0.5, 0.75 and 0.95 are shown in Figure 5, in which the vortex structures are identified by the isosurface of the Q-criterion. At the beginning of the downstroke, the hindwings exhibit rapid pronation and downward flapping, and unstable vortices were observed on the surface of the hindwings as a result of their generation during the previous upstroke and the result of shedding into the downwash flow. In the first half of the downstroke, both LEV and TEV were developed. At the middle of the downstroke (*t*/*T* = 0.3), the lift reached the maximum, and the LEV was fully developed and the strength of the vortex was enlarged from the wing base to the wing tip. This may provide another mechanism for enhancing lift generation. At the end of the downstroke (*t*/*T* = 0.5), the LEV was detached from the hindwings and shed to downstream. A strong wing-tip vortex developed from the hindwing, and the vortex rings elongated. The continued upstroke of the hindwings generates 3D vortex rings, including a leading-edge vortex (LEV), a trailing-edge vortex (TEV) and a wing-tip vortex (TV). At the mid-upstroke (*t*/*T* = 0.75), the spiral LEV can be clearly observed on the lower surface of the hindwings, which is a striking phenomenon in 3D flapping-wing motion. The LEV generated during the upstroke is stable, and most of it attaches to the surface of the hindwings until the mid-upstroke motion. As the wing approached the end of the upstroke, at *t*/*T* = 0.95, a strong LEV was generated on the ventral side of the surface. An unstable vortex ring region was found in the region near the wing tip of the hindwings due to the rapid rotational motion during the transition phase from the upstroke to the downstroke.

Aerodynamic pressure is non-dimensionalized by wing load, which is the ratio of *C. buqueti’s* weight to the total surface area of the hindwings. As shown in Figure 6, before the hindwings approached the first half of the downstroke, at *t*/*T* = 0.15, a negative pressure region was generated on the upper surface of the hindwings when LEV was strengthened. The positive pressure region was created on the bottom surface due to the downward motion. At this time, the vertical force and the horizontal force reached the maximum value. The vertical force gradually decreased before the end of the downstroke, and it had small amplitude changes within a short period of time when the downstroke reached *t*/*T* = 0.35–0.5. The vertical force transitioned to a negative value on the subsequent downstroke motion. The spiral LEV was forced to attach on the dorsal side due to spanwise flow. In other words, the separation and shedding process of LEV was delayed in the 3D flapping wings; for example, at *t*/*T* = 0.35, there was still a spiral LEV attached to the hindwings. Similarly to the downstroke, the upstroke motion had a ring-shaped vortex wake structure, but it was not as strong as the vortex of the downstroke. After the middle of the upstroke, at *t*/*T* = 0.8, a positive pressure area was formed on the dorsal side of the hindwings, and a negative pressure area was formed on the ventral side of the hindwings.

The aerodynamic forces acting on the hindwings can be calculated by integrating the pressure and shear stress on the dorsal side and ventral side surfaces of the hindwing model. Then, vertical and horizontal forces can easily be computed by converting the total force into the horizontal direction and the vertical direction. The non-dimensional lift coefficient (*C_L_*) and drag coefficient (*C_D_*) used to represent vertical and horizontal forces are defined as
(6)$$C_{L}=\frac{F_{L}}{0.5\rho U^{2}S_{W}}$$
(7)$$C_{D}=\frac{F_{D}}{0.5\rho U^{2}S_{W}}$$
where *F_L_* and *F_D_* represent the lift force and drag force, respectively. *S*_W_ is the projected area of a single hindwing in the direction of motion, which is 0.448 × 10^−3^ m^2^.

As shown in Figure 7, there are two *C_L_* peaks in one wing beat cycle: one in the first half of the downstroke cycle and the other in the second half of the upstroke cycle. The first peak is significantly larger than the second peak. The peak is contributed to by the leading-edge vortices and the wing-tip vortices of the flapping hindwings. The ability of *C. buqueti* to obtain stable lifts and thrusts during flight depends on the shorter wing-beating time. In addition, the thrust force for a longer time period occurs during the upstroke. The lift force is generated in the short time before the end of the downstroke and the time before the middle of the upstroke. A large amount of lift force is produced during the downstroke, which accounts for about 61% of the total lift force generated by the entire wing beat cycle. The averaged vertical force during one stroke cycle is 0.09 N. The time-averaged horizontal force during the entire stroke cycle is −0.022 N. Considering that the weight of the *C. buqueti* is 4.02 g, the vertical force computed by CFD simulations is 2.15 times that of *C. buqueti*.

### 3.4. Asymmetrical Bending

The asymmetrical bending of the hindwings subjected to aerodynamic pressure loads plays an important role in evaluating aerodynamic effects. The pressure load is equal to the ratio of the average aerodynamic force calculated by CFD to the area of the flying wing. The pressure load is defined by *q* = *F*/2*A*, where *F* is the averaged vertical force (0.09 N) and *A* is the area of the hindwing measured by Unigraphics NX software (0.4833 × 10^−3^ m^2^). A pressure of 1.03 × 10^−4^ MPa was applied from the ventral and dorsal sides, respectively, and the simulation results are shown in Figure 8. The deflections induced by pressure loads from the ventral and dorsal sides gradually increased from the wing root to the wing tip. The largest structural deformation occurred near the wing tip, which was 1.3428 mm and 1.3391 mm. The maximum stresses generated by the pressure load from the ventral side and the dorsal side were 27.731 MPa and 27.806 MPa, respectively. The maximum stress is much smaller than the material properties of the hindwings, which means that the stress concentration and fatigue failure can be avoided during the flapping of the hindwings. These results demonstrate that the bending deformation of the hindwings loaded from the ventral side is larger than that of the hindwings loaded from the dorsal side, which indicates that the stiffening effect plays an important role in the asymmetrical bending of the hindwings of *C. buqueti*.

## 4. Discussion

### 4.1. Wing Kinematics Analysis

A pair of hindwings of *C. buqueti* have the same flapping-wing trajectories and share a stroke plane. The intersection of the flapping trajectories of the hindwings occurs when pronation and supination are imminent [21]. This intersection determines whether the flapping pattern of insect wings is a simple figure-eight trajectory or a complex double figure-eight trajectory [40]. The most likely explanation for the different wing beat trajectories during flight is that insects with different aerodynamic properties employ different flapping-wing patterns. Insects often use different aerodynamic mechanisms to fly during flapping-wing flight, and they can autonomously choose different mechanisms for the same flapping-wing motion [40,41]. The large pressure difference between the ventral side and the dorsal side of the hindwings of the *C. buqueti* during the downstroke forces the formation of an umbrella-shaped deformation with respect to the hindwings (Figure 3), which can effectively keep the insect hovering in the air [42]. There are no obvious umbrella-like deformations with respect to the hindwings of *C. buqueti* during the upstroke, which may be related to the reduced drag [42].

The effective processing of the measured kinematic data is key to constructing an accurate mathematical model, which can produce numerical simulation results that are closer to the flight conditions of real insects. The literature reports that there is a significant difference in the force between real and harmonic functions in a fruit model [19], which cannot truly reveal the kinematics of fruit flight. To evaluate the aerodynamic performance of real flapping-wing kinematics in flight, high-order sinusoidal sequences are used in this paper, where the Fourier order and its coefficients determine the accuracy of the results. The high-speed camera captures the body angle of *C. buqueti* during multiple attitude adjustments in the tethered flight, and the angle is recorded as zero degrees relative to the horizontal direction, which may be a unique feature of *C. buqueti’s* hovering flight. It was also reported that *C. buqueti* generates positive lift forces during the flapping motion, and the average lift force is higher than its own body weight. In addition, *C. buqueti* can use increased thrust force to make up for the lack of flight lift force [21].

### 4.2. Flapping Aerodynamic Characteristics

The flow field around the wings of the *Drosophila* robot [43] was simulated to verify the accuracy of the CFD solver used in our previous work [37]. The flapping-wing kinematic data were derived from the experimental results. The generation of beneficial aerodynamic forces is mainly contributed to by the downstroke of the hindwings and has also been reported in other insect flights [4,8,44,45]. The hindwings of *C. buqueti* generate drag during the downstroke and thrust by pushing the hindwings backwards during the upstroke, which is generated by redirecting the lift force through the forward tilting motion of the hindwing’s stroke plane [4]. It was found that the aerodynamic performance of the wing-beat stage of the hindwings may be different in the same wing section; for example, high lift can be general in the middle of the downstroke and thrust can be generated near the middle of the upstroke (Figure 6). The ventral side of the hindwings accumulates a certain number of LEVs before the end of protonation, which will be beneficial for generating lift and thrust during the subsequent downstroke [46]. The hindwings during supination gain an increase in drag during the transition from the downstroke to the upstroke.

The low-pressure region on the dorsal surface (Figure 9a) can be clearly seen from the streamlines near the leading edge and the wing tip of the hindwings. The spiral streamlines grow in the wing’s spanwise direction from the wing root to the mid-wing area. The vortices emanating from the wing tip area of the hindwings contribute to the downwash flow. As shown in Figure 5b, the TV and TEV form a contralateral vortex ring on both sides of the body of *C. buqueti* so that a double-vortex ring structure forms in the near wake. The dynamic stall of the LEV on the hindwings is stabilized by the strong spanwise flow [47]. As shown in Figure 6 and Figure 9, the vortex and pressure distributions at *t*/*T* = 0.5 demonstrate the clear role of spanwise flows during flapping.

### 4.3. Asymmetrical Bending Analysis

In order to verify the accuracy of the inversely reconstructed hindwings, two methods were used for the analysis. One method compares the simulation results of CFD with the results measured by the motion capture system. As shown in Figure 10a, which compares the vertical force obtained by the CFD simulation with the vertical force obtained by the motion capture system experiment [21], it was found that the two curves are in good agreement, which indicates that the 3D model of the reverse reconstruction accurately approximates real hindwings. The other is to apply the displacement at the MJ and obtain the change characteristics of the reaction force by using the bending test and the finite element simulation methods, respectively. The thickness of the 3D model is 0.1 mm, which can be 3D printed as a physical prototype. Photosensitive resin materials are selected for 3D printing, and photosensitive resin is also selected for the material properties of the finite element simulation. Figure 10d shows the force response of the MJ of the hindwings with a displacement of 1.2 mm using finite element simulations and the bending experiment, respectively. The deviation of the experimental and simulation results is acceptable, which also shows that the bionic model obtained by reverse modeling technologies is the same as the real hindwing model.

For the structural characteristics of the hindwings, the wing’s veins act as a stiffener so that the wing’s membrane can bear web-shear forces with pure tension. Therefore, when the hindwings are loaded and bent, the out-of-plane stiffness of the structure will be significantly affected by the in-plane stress state of the airfoil, which allows improved bearing capacities for the wing membrane under tension [33]. The finite element simulation indicates that the hindwings are more rigid when the pressure load is applied to the dorsal side than when it is applied to the ventral side. A plausible explanation for this phenomenon is that the chordwise camber of the hindwings increases the rigidity when the pressure load is applied from the ventral surface, while the spanwise camber increases the rigidity when the pressure load is applied from the dorsal surface. In addition, from the morphology of the hindwing [39,48] compared with the ventral side, there are convex veins on the dorsal side that may also result in more rigid hindwings when loaded from the dorsal side. In other words, the hindwings are more flexible during the downstroke, which can generate better aerodynamic performance, which is consistent with the high proportion of vertical forces obtained by the downstroke of the hindwings shown by the results of the CFD simulation.

## 5. Conclusions

The flight mechanism of *C. buqueti* in hovering flight was explored by using numerical simulations. The real 3D model of the specimen was obtained by using three-dimensional scanning technology and reverse surface reconstruction technology. The flapping kinematics of the hindwings were measured using high-speed cameras and reconstructed using a modified direct linear transformation algorithm. The finite element simulation of the pressure load applied from the dorsal and ventral sides shows that the hindwings exhibit asymmetric bending, which may be determined by both the camber’s morphology and the effect of hindwings stiffening under stress. The hindwings generate thrust during upstroke motions and experience drag during downstroke motions, and approximately 61% of the total lift is generated during the downstroke, which is a result of the asymmetric bending of the hindwings. In addition, two more intuitive methods were used to verify the accuracy of the inverse modeling technique applied to the hindwings. This research will provide a theoretical reference for improvement in the aerodynamic performance of flapping-wing MAVs.

## Figures and Tables

**Figure 1 micromachines-13-01995-f001:**
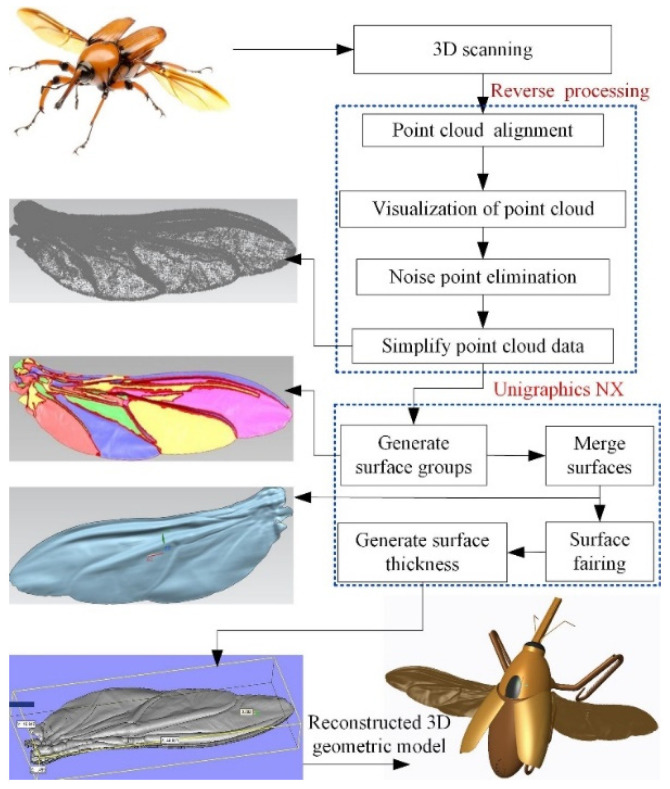
Reverse reconstruction of the specimen.

**Figure 2 micromachines-13-01995-f002:**
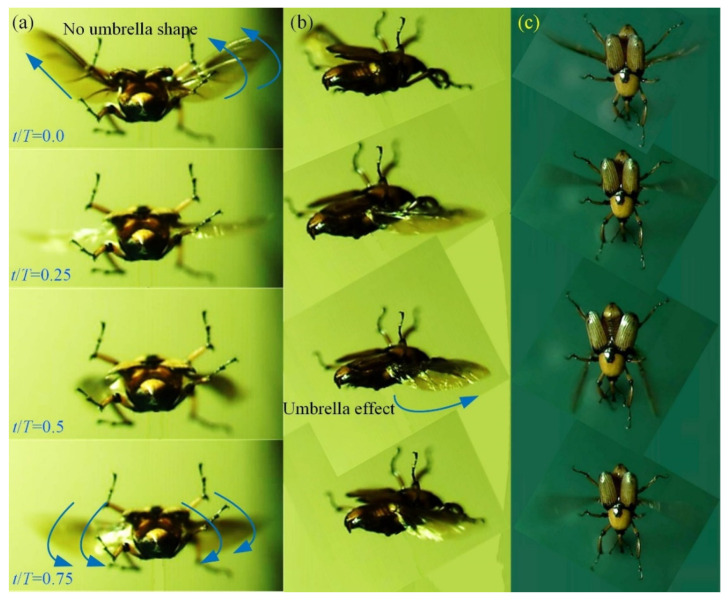
Four stages of the flapping wing cycle. (**a**) Front view. (**b**) Right view. (**c**) Aerial view.

**Figure 3 micromachines-13-01995-f003:**
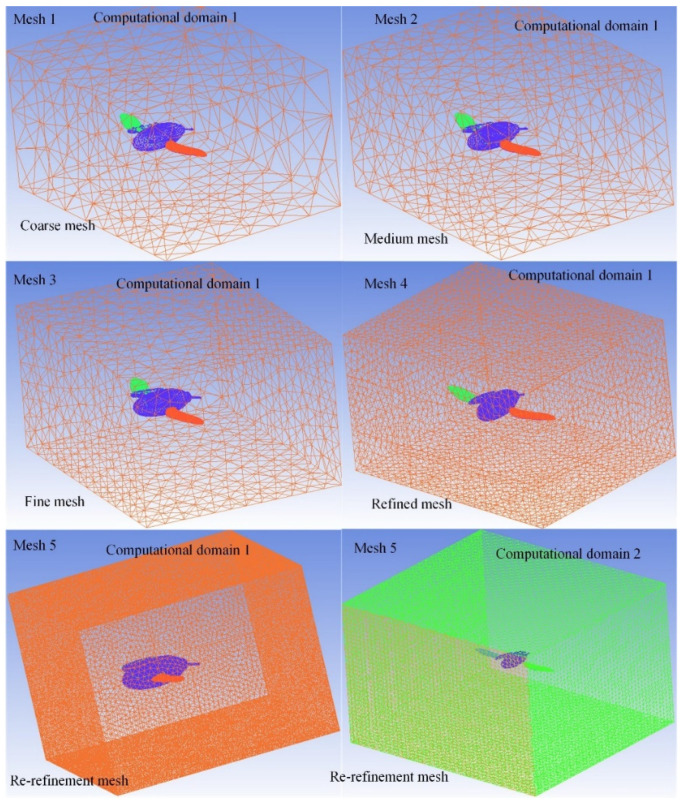
Demonstration of five different mesh systems and two different computational domains.

**Figure 4 micromachines-13-01995-f004:**
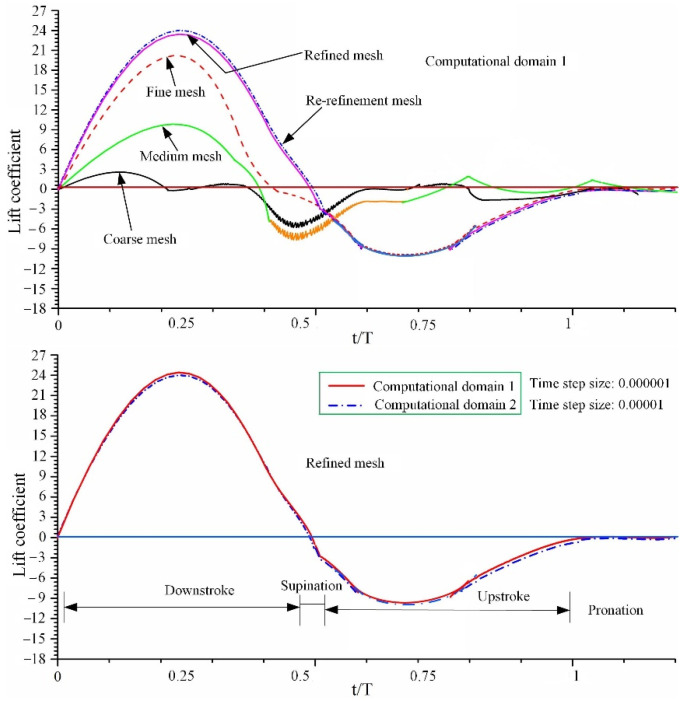
Comparison of lift coefficient curves between four different mesh qualities and two different computational domains.

**Figure 5 micromachines-13-01995-f005:**
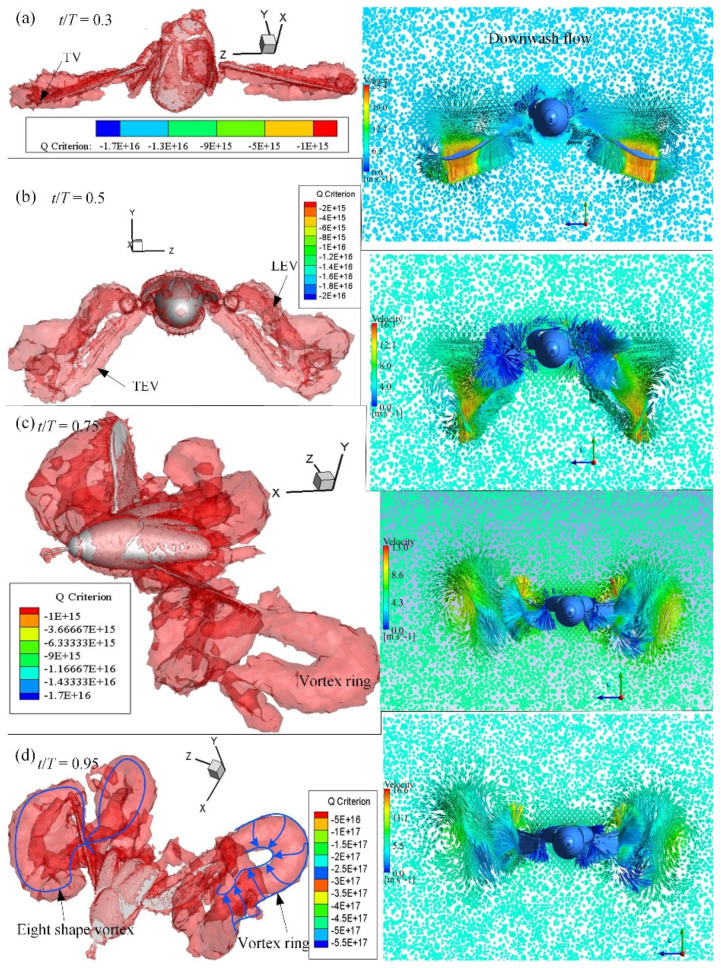
Isometric views of the Q-criterion (**left column**) and downwash velocity (**right column**) on the wing’s surface at some typical time steps.

**Figure 6 micromachines-13-01995-f006:**
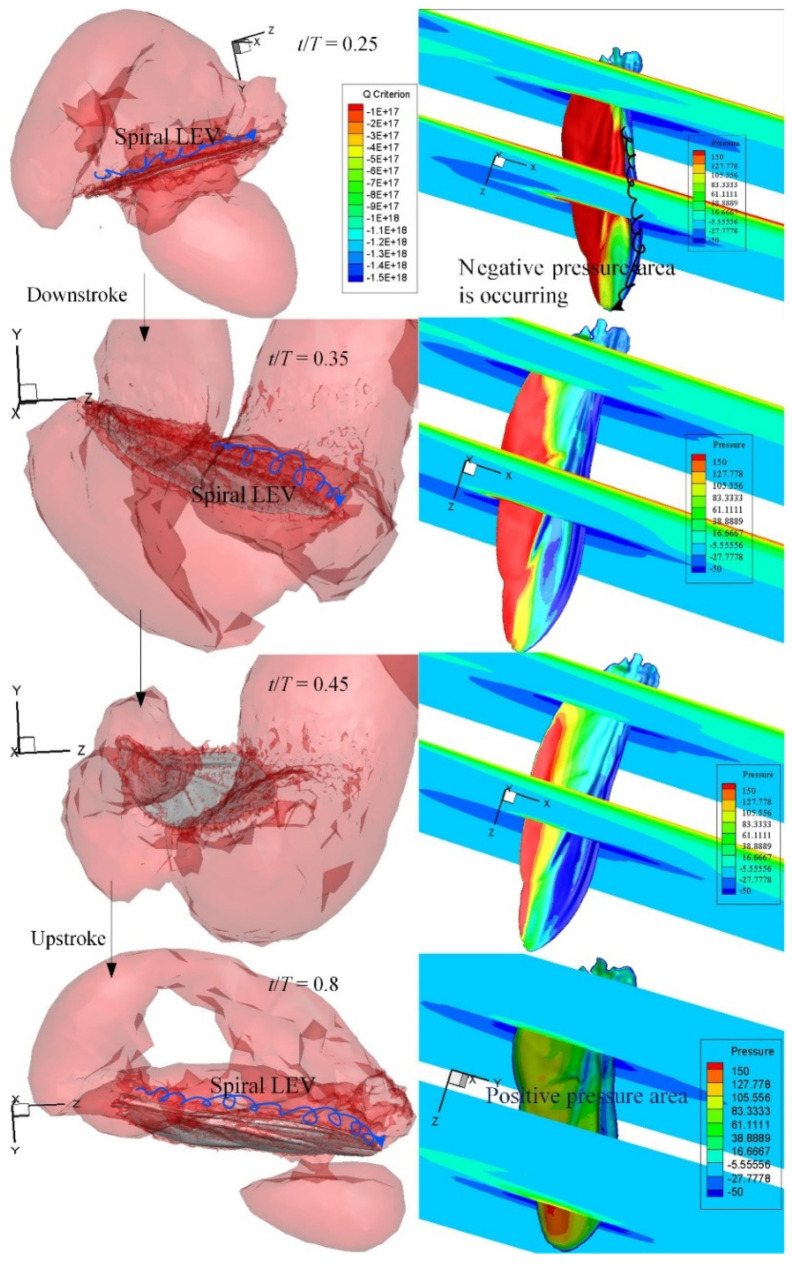
Effect of hindwing morphology on vortices (**left column**) and pressure distribution (**right column**) around the wing surface at some typical time steps.

**Figure 7 micromachines-13-01995-f007:**
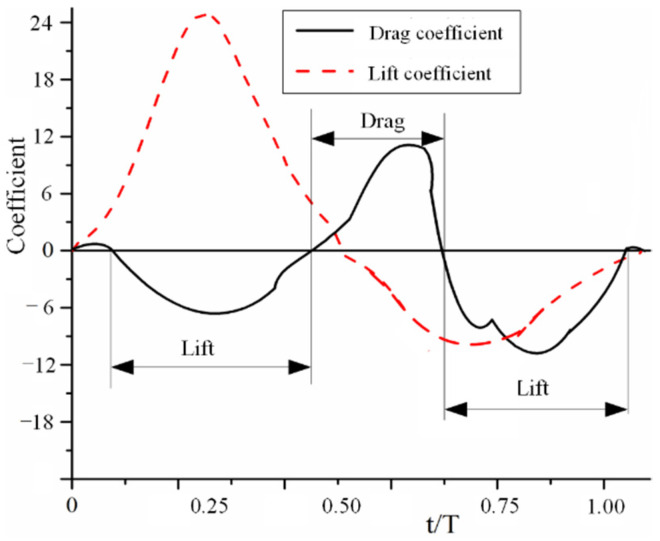
Force change of *C_L_* and *C_D_* with the time for the flapping of hindwings.

**Figure 8 micromachines-13-01995-f008:**
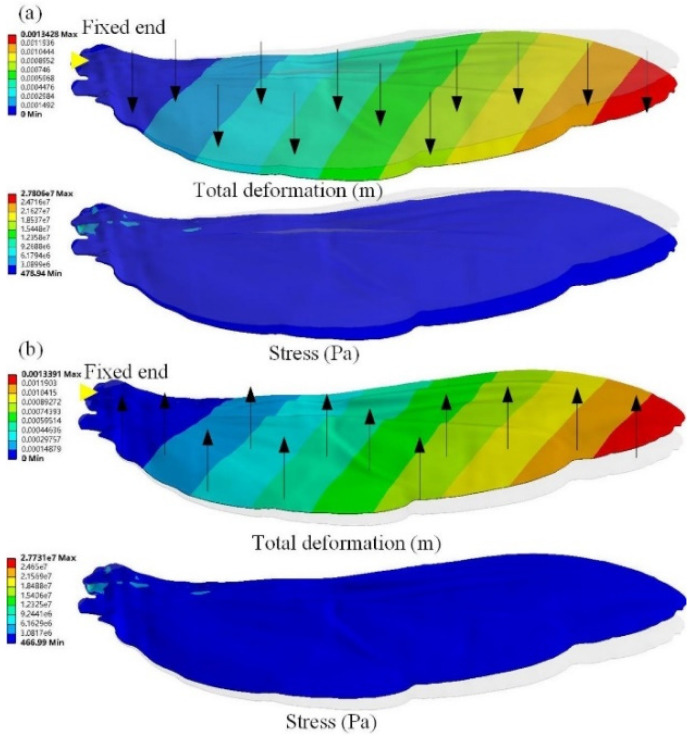
Finite element simulation results with pressure loads from ventral and dorsal sides. (**a**) The pressure load is applied to the dorsal side; (**b**) the pressure load is applied to the ventral side.

**Figure 9 micromachines-13-01995-f009:**
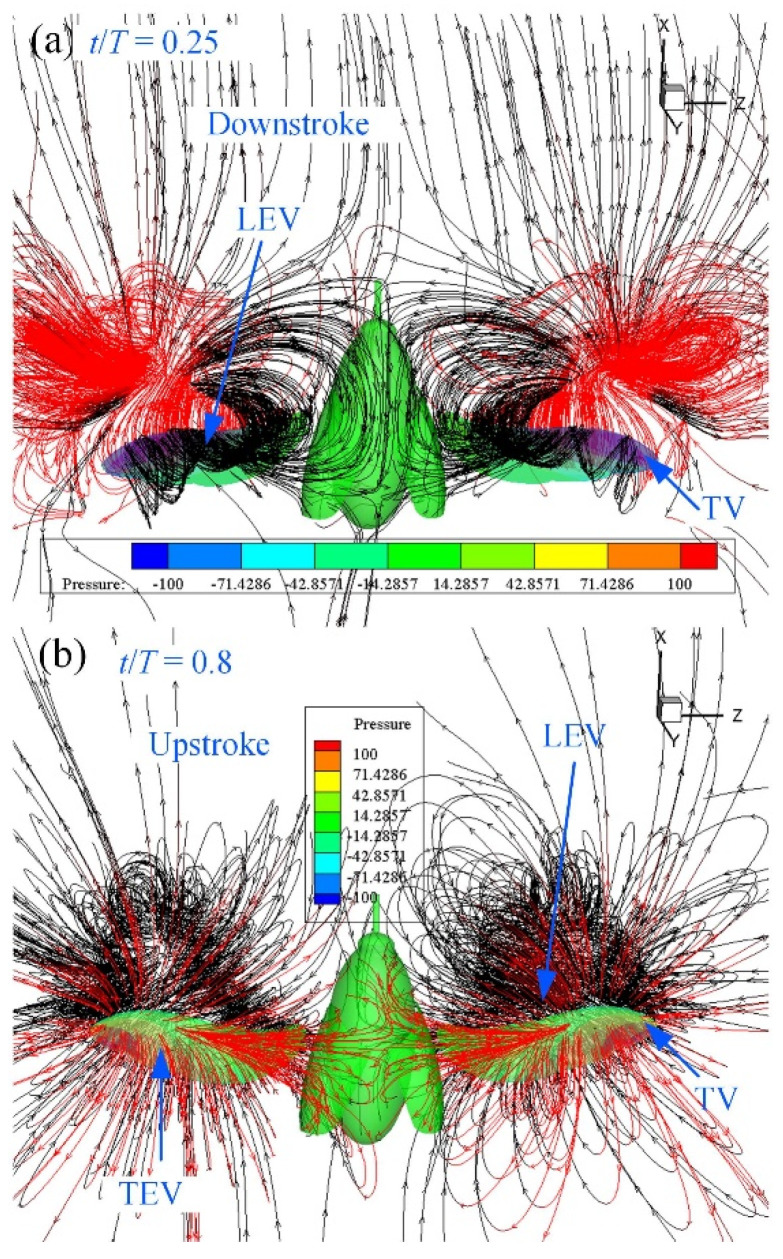
Streamlines around the hindwings at mid-downstroke and near mid-upstroke. (**a**) The LEV and the TV at mid-downstroke. (**b**) The LEV, TEV and the TV at near mid-upstroke.

**Figure 10 micromachines-13-01995-f010:**
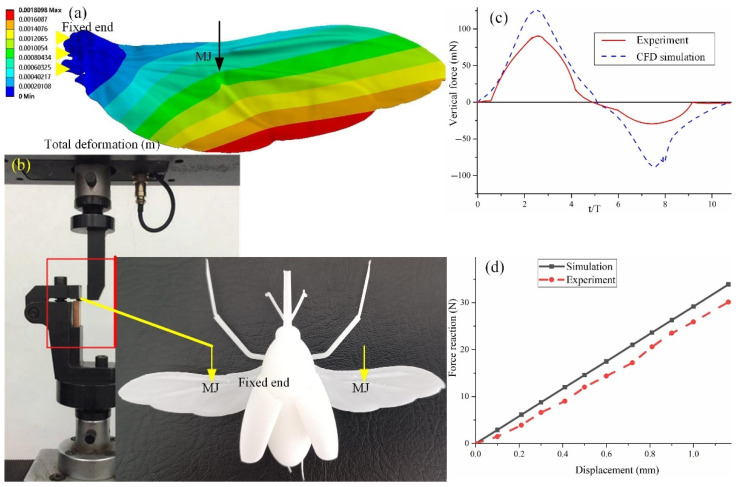
Comparison of simulation results with experimental results. (**a**) Deformation simulation for a 1.2 mm displacement at point MJ. (**b**) Bending experiment for a 1.2 mm displacement at point MJ. (**c**) Comparison of CFD simulation and experimental results. (**d**) Comparison of results from finite element simulations and bending experiments.

## Data Availability

The data presented in this study are available upon request from the corresponding author.

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
