# Peer review of "Numerical Simulations of the Effect of the Asymmetrical Bending of the Hindwings of a Hovering C. buqueti Bamboo Weevil with Respect to the Aerodynamic Characteristics"

_micromachines, 2022, doi:10.3390/mi13111995_

Round 1

Reviewer 1 Report

The manuscript entitled “Numerical simulation of the effect of asymmetrical bending of the hind wings of a hovering C. buqueti bamboo weevil on the aerodynamic characteristics” by Xin Li. has been reviewed. The authors investigated the aerodynamic performance of the hind wings of the insect with asymmetric bending effects in hovering flights. It is a nice piece of work that will be of good interest to the Micro Air Vehicle community. Therefore, I only recommend some minor revisions. I recommend this paper be accepted after the following minor concerns are addressed.

The introduction needs to be revised extensively to reflect the major findings from previous investigations, and more details about the results of the previous finding need to be mentioned.

Line 97 explains more about the accuracy of the reconstructed model and how it was verified.

Line 99 did the treatment process affect the corrugation patterns of the wing and what is the difference between the live and dead insect’s wings?

There are many grammatical errors.

Author Response

Replies to reviewer 1

  1. The introduction needs to be revised extensively to reflect the major findings from previous investigations, and more details about the results of the previous finding need to be mentioned.

Answer: The introduction has been partially modified (marked in red). There are few papers on the asymmetric bending of insect wings and its effect on aerodynamic forces.

2 Line 99 did the treatment process affect the corrugation patterns of the wing and what is the difference between the live and dead insect’s wings?

Answer: The treatment process does not affect the corrugation patterns of the wing. There is no difference between the wings of living and dead insects. The wings of dead insects were removed for convenience of preservation, 3D scanner and scanning electron microscope. Live insects are used for flight kinematics testing.

  1. There are many grammatical errors.

Answer: Many grammatical errors have been corrected by the paper editing service (marked in blue).

I look forward to working with you to move this manuscript closer to publication in the Micromachines.

Thank you very much.

Wish best wishes,

Yours sincerely,

Dr. Li

Reviewer 2 Report

The simulation results of the flight mechanism of the C. buqueti in hovering flight are very interesting and have meaningful findings. 

The results are sufficient and I think suitable to publish directly.

Author Response

Replies to reviewer 2

Many grammatical errors have been corrected by the paper editing service (marked in blue).

I look forward to working with you to move this manuscript closer to publication in the Micromachines.

Thank you very much.

With best wishes,

Yours sincerely,

Dr. Li